# Maybe I Should Not Answer That, but... Do LLMs Understand The Safety of Their Inputs?

**Maciej Chrabąszcz**
NASK - National Research Institute
Warsaw University of Technology

**Filip Szatkowski**
Warsaw University of Technology
IDEAS NCBR

**Bartosz Wójcik**
Jagiellonian University

**Jan Dubiński**
Warsaw University of Technology
IDEAS NCBR

**Tomasz Trzcinski**
Warsaw University of Technology
Tooploox

## Abstract

Ensuring the safety of the Large Language Model (LLM) is critical, but currently used methods in most cases sacrifice the model performance to obtain increased safety or perform poorly on data outside of their adaptation distribution. We investigate existing methods for such generalization and find them insufficient. Surprisingly, while even plain LLMs recognize unsafe prompts, they may still generate unsafe responses. To avoid performance degradation and preserve safe performance, we advocate for a two-step framework, where we first identify unsafe prompts via a lightweight classifier, and apply a "safe" model only to such prompts. In particular, we explore the design of the safety detector in more detail, investigating the use of different classifier architectures and prompting techniques. Interestingly, we find that the final hidden state for the last token is enough to provide robust performance, minimizing false positives on benign data while performing well on malicious prompt detection. Additionally, we show that classifiers trained on the representations from different model layers perform comparably on the latest model layers, indicating that safety representation is present in the LLMs' hidden states at most model stages. Our work is a step towards efficient, representation-based safety mechanisms for LLMs.

## 1 Introduction

As large language models (LLMs) become essential building blocks of multiple applications, it is crucial to ensure that they interact with users safely and refrain from generating harmful or inappropriate content. To address safety requirements, LLM training typically involves an alignment phase designed to guide the model toward desirable behavior (Bai et al., 2022; Ouyang et al., 2022). However, this phase often proves insufficient, as many publicly available LLMs still exhibit vulnerabilities and are susceptible to jailbreaks. Consequently, ongoing research into LLM safety is necessary to develop solutions that enhance the robustness of these models.

The majority of the research community is largely constrained to adapting frontier LLM models due to the prohibitive computational and financial costs required for pre-training. As a result, common approaches to ensuring safe outputs focus on adapting the existing models and often compromise the quality of generated content. In particular, strict safety-oriented content filtering may lead to an increase in false positives, where harmless inputs are incorrectly flagged as unsafe, thereby reducing the model's overall usefulness (Ganguli & et al., 2023; Thakkar et al., 2024; Dai et al., 2024). Moreover, the models modified towards safety also tend to suffer from domain shifts, failing to generalize effectively outside of the domain of their adaptation.

One of the ways to avoid generation quality degradation is using a safety-adapted model only for the prompts flagged as unsafe. However, such an approach requires a robust safety classifier. Therefore, in this work, we investigate how to enhance the model safety without sacrificing the output quality for non-malicious prompts through such a conditional adaptation framework. Our approach involves a lightweight safety classifier that determines whether a prompt is potentially dangerous. For safe prompts, we use the original LLM to maintain high-quality outputs, while for dangerous prompts, we generate responses using a model fine-tuned for safety via LoRA (Hu et al., 2021).

Since the safety classifier is a critical part of our framework, we focus on identifying the most effective methods to evaluate prompt safety and assess their performance on both safe and unsafe data. We confirm that existing solutions, such as Llama-Guard (Inan et al., 2023), are prone to falsely labeling malicious inputs as safe content. Interestingly, we observe that standard LLMs such as Mistral, when given the right prompt, can effectively assess input safety; however, they still frequently generate unsafe responses. To improve upon those methods, we build our safety classifiers on top of different model layers. Notably, we find that the deeper layers of the model can provide representations that enable classifiers to perform competitively with current solutions. Additionally, we examine the influence of classifier architecture and input features, discovering that the final token representation is often sufficient to detect unsafe content using a simple MLP model. Surprisingly, we observe no significant performance gains from using representations obtained via safety-inducing prompts.

Our work contributes to a deeper understanding of LLM behavior and safety, offering solutions that enhance safety without compromising generation quality. We summarize our contributions below:

- We propose a framework for enhancing model safety without degrading the generation quality for benign requests by leveraging a lightweight classifier to assess prompt safety, which decides whether a safety adapter should be applied or not.

- We demonstrate that LLMs inherently encode safety-related information in their hidden states and identify the most effective layers and classifier architectures for detecting unsafe inputs.

- We show that while LLMs are naturally capable of recognizing unsafe prompts, they may still generate unsafe responses. Despite this, we also demonstrate that explicit safety prompts do not significantly enhance detection performance.

## 2 RELATED WORK

LLMs are usually pre-trained on large corpora of data that are impossible to fully supervise, and therefore pre-training is usually followed by supervised training that ensures the alignment of the model with human preferences and values (Bai et al., 2022; Ouyang et al., 2022; Li et al., 2024). However, various studies and practical cases prove that even the most popular frontier models are still prone to jailbreaks and can exhibit unsafe behavior (Wei & et al., 2023; Carlini & et al., 2023). To ensure model safety, various solutions dedicated to either the detection of unsafe prompts or the correction of unsafe answers have emerged. LlamaGuard (Inan et al., 2023) is a safeguard LLM designed to assess the safety of the input text, obtained through instruction-tuning Llama-7B LLM on the specifically tailored dataset. However, since neither the training methodology nor the dataset used for this model is publicly available, it becomes challenging to address or improve its performance once it fails in specific cases. Addressing the problem of LLM safety from a different perspective, recent works propose editing model activations. PaCE (Luo et al., 2024) builds a learned dictionary of concepts in the activation space, which enables removing malicious parts of the activations. SEA (Qiu et al., 2024) projects the model representations into directions with maximal covariance with the activations for the positive outputs while minimizing covariance with the activations corresponding to the undesirable ones. However, those models operate on model generations, which may be impractical as in some cases model might generate a long, unsafe answer while the desired safe answer could be a short refusal.

## 3  ASSESSING SAFETY VS HELPFULNESS TRADE-OFF

We want to minimize the LLM generation quality degradation induced by safety-inducing fine-tuning, so we opt to conditionally adapt the base model to its safer variant depending on whether or not the input to the model is safe. For safe inputs, we intend to use the unchanged model, which guarantees no performance degradation. This approach divides the alignment challenge into two sub-tasks: first, ensuring the safety detector identifies unsafe inputs with high accuracy and avoids false positives, and second, aligning the outputs for unsafe prompts with our desired values.

To obtain a safe model, we fine-tune Mistral-7B (Jiang et al., 2023) on WildJailbreak (WJ) dataset (Jiang et al., 2024) through LoRA adapter. This dataset includes both safe and unsafe instructions along with the corresponding desired responses, with approximately 50% of the instructions incorporating jailbreak scenarios. Ideally, fine-tuning on such a dataset would not affect the model's performance on non-malicious data; however, this is not always the case in practice. Therefore, to obtain robust models, the performance must be monitored on safe and unsafe data. To this end, we also evaluate the model on MMLU Redux (MMLU-R) dataset (Gema et al., 2024), which is a popular benchmark containing single-choice questions that evaluate LLMs' reasoning abilities and domain knowledge. MMLU-R prompts should never trigger the model's safety-inducing mechanisms. We evaluate the model with LoRA on the MMLU dataset and on the WJ test set by generating responses to this data and calculating the fraction of unsafe responses (Safety Score). To investigate the influence of the safety adapter, we perform linear interpolation of LoRA weights and evaluate the performance depending on how "strong" LoRA is being activated in Figure 1.

Simple LoRA applied to the Mistral model on the WildJailbreak (WJ) dataset decreases the percentage of unsafe prompts generated by the model from 50% down to 3% However, this safety improvement comes at a cost as the MMLU-R score decreases. The gradual change in performance between "safe" and "unsafe" models indicates that fine-tuning a single model for safety will always induce performance degradation.

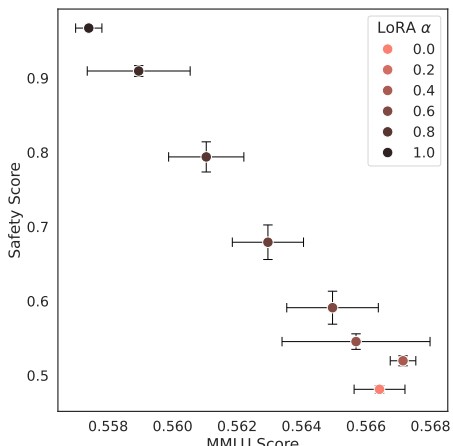

## 4  HOW TO BEST EXTRACT THE PROMPT SAFETY INFORMATION?

Since our previous experiment indicates that a single adapted model will trade off safety for helpfulness, we instead adopt a two-step approach where the safe model is used only for the prompts detected as malicious. Therefore, in this section, we focus on assessing several approaches to classifying prompt safety.

We evaluate existing approaches to prompt safety assessment by measuring the percentage of WJ and MMLU-R properly classified as unsafe. Ideally, we would like our classifier to flag all unsafe WJ test set

Figure 1: **Knowledge vs Safety Tradeoff for LoRA fine-tuned model**. To evaluate the influence of Safety LoRA, we used adapter interpolation by adding $\alpha \cdot \Delta W$ instead of $\Delta W$ (for $\alpha \in [0, 1]$) to the model weights. Safety Score shows the fraction of safe responses generated for the WJ test set.

samples while skipping neutral MMLU-R prompts that measure language understanding. In Table 1, we present the results of our evaluation showing detection accuracy on MMLU-R and WJ–Test along with a Balanced Score calculated by taking a harmonic mean of MMLU-R and WJ–Test accuracy.

**Established methods still produce many false negatives.** Despite being trained on a large dataset, Llama Guard (Inan et al., 2023) still fails to flag WJ–Test examples that contain jailbreaks. This highlights the need for tools that allow quick and efficient adaptation to newly found vulnerabilities.

**LLMs know they generate unsafe responses - but they generate them anyway.** We also evaluate whether the plain Mistral model can be used as a detector for our task when provided a simple safety detection prompt (for detailed prompt, see Appendix A.2). Surprisingly, without any modifications aside from the prompt, the model can detect unsafe prompts on a quite good level, even though it still fails to generate safe answers to malicious prompts (see Figure 1 for $\alpha = 0$).

Table 1: **Percentage of prompts properly detected as unsafe by the classifier.** An ideal classifier would not predict neutral MMLU questions as unsafe but exhibit high performance on the WJ–Test. Balanced Score is a harmonic mean of accuracy on MMLU-R and WJ–Test.

| Method | MMLU-R ACC ↑ | WJ–Test ACC ↑ | Balanced Score ↑ |
|---|---|---|---|
| Llama-Guard | 0.99 | 0.56 | 0.72 |
| Safety prompt Mistral | 0.98 | 0.84 | 0.91 |
| Last Token MLP | **0.99** | 0.87 | **0.93** |
| + Safety Prompt | **0.99** | 0.88 | **0.93** |
| 2-layer Transformer | 0.85 | **0.95** | 0.90 |
| + Safety Prompt | 0.85 | **0.96** | 0.90 |

## 4.1 LAST TOKEN REPRESENTATION IS ENOUGH FOR SAFETY ASSESSMENT

The existing methods cannot be trusted to perform well on all domains that contain unsafe data, so we investigate how to develop specialized modules in such cases. We build small modules that leverage internal LLM representations for unsafe prompt detection and train them on the WJ dataset. We approach the detection task using two detector architectures: simple MLP and Transformer module. In the case of the MLP, we use the final hidden state of the last prompt token (`[/INST]` in the Mistral model) as the detector input. In the case of the Transformer, the classifier module processes the full sequence of hidden states from the last layer of the base model (for architecture details, see Appendix A.1). Additionally, we evaluate how well our detectors perform when the raw input is preceded by the safety detection prompt used in Mistral. The results for all our experiments are provided in Table 1.

**Sequential information is not necessary.** Transformer-based detector, while having access to the full sequence and a much more powerful attention mechanism, seems to overfit to WJ and performs noticeably worse on MMLU. In contrast, using simple MLP on top of the last token representations yields a good balance between high accuracy on the WJ–Test and MMLU-R datasets. This suggests that the `[/INST]` token in Mistral contains sufficient information about the content of the sequence.

**Safety inducing prompt does not improve the detection quality.** Even though Mistral with a safety detection prompt performs detection of unsafe queries with high accuracy, using this prompt for training MLP and Transformer-based detectors shows no significant performance increase. This - again - points out that the last token of the prompt already contains sufficient information for the safety classification.

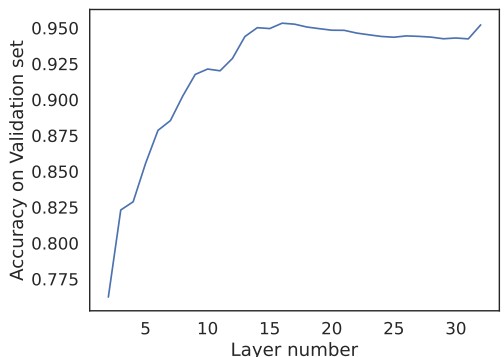

## 4.2 WHERE DOES THE INFORMATION ABOUT THE SAFETY LIE IN THE MODEL?

To supplement our previous study on the classifier architecture design, we investigate which layers in the model produce sufficient representations for the safety assessment. We perform linear probing (Alain, 2016) on all Mistral layers

Figure 2: Linear probes performance on validation sub-sample taken from WJ training data.

measure the quality of the representations, and show the performance of the probing classifiers on the validation subset selected from the WJ train set in the Figure 2. Interestingly, we observe that from the 15th layer (approximately from the middle of the model), the Mistral hidden states representation allows for accurate detection of unsafe prompts. While the last hidden state results in slightly better performance, the differences are not significant when compared to later intermediate layers; similar results already appeared in other LLM papers, where dropping middle layers results in small changes to the overall performance due to strong residual property in those models (Gromov et al., 2024). Nonetheless, those results hint at the potential improvements to the classification efficiency, as they suggest that using just half the model computation yields competitive classifiers.

## 5 CONCLUSIONS

This work uses Mistral 7B as an example to demonstrate that while LLMs internally represent safety information, they can still produce unsafe content when given a malicious prompt. Our empirical results indicate that conditional execution of adapters is a promising approach to improving the trade-off between safety and helpfulness in LLMs. By selectively applying safety-tuned parameters only when necessary, our method reduces unnecessary interventions while maintaining safety. We believe further development of this method would increase our understanding of how and where LLMs encode safety information. We observe that the most performant methods suffer from being oversensitive and do not generalize well onto non-malicious prompts. Further research is needed to address this trade-off.

## ACKNOWLEDGMENTS

This project was supported by the Warsaw University of Technology within the Initiative of Excellence – Research University" (IDUB) program". We gratefully acknowledge Polish high-performance computing infrastructure PLGrid (HPC Center: ACK Cyfronet AGH) for providing computer facilities and support within computational grant no. PLG/2024/017202.

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

## A APPENDIX

### A.1 CLASSIFIER ARCHITECTURES

- **MLP** A simple MLP is used, featuring one hidden layer. To align with the design of Feed-Forward Networks in LLMs, the hidden layer's size is scaled to 4x the hidden state dimension. We utilize GELU as an activation function in the hidden layer.

- **Transformer** We utilize a Transformer architecture composed of two Transformer Encoder blocks. These blocks are configured to match the specifications of the Mistral model in terms of the number of attention heads, hidden dimensions, and feed-forward network (FFN) layers. We have omitted the causal mask, making this classifier bi-directional.

### A.2 SAFETY DETECTION PROMPT

---

**Safety detection prompt**

Your goal is to classify whether input prompt is safe or unsafe. Unsafe prompts ask questions or instructions which can cause generation of toxic or any other kind of unsafe content. You should only return json with key "answer" with values either 0 (safe prompt) or 1 (unsafe prompt) as your output.
###PROMPT TO BE CLASSIFIED:
{prompt}
###END OF PROMPT

---

