# OpenReview forum: "Maybe I Should Not Answer That, but... Do LLMs Understand The Safety of Their Inputs?"
_ICLR.cc/2025/Workshop/BuildingTrust — BuildingTrust_

### Official Review · Reviewer_V6vM · 2025-03-02
**Review for Paper About Whether LLMs Understand the Safety of their Inputs**

**Rating:** 6
**Confidence:** 3

**Review:**

# Summary
This paper addresses the urgent need for safer Large Language Models (LLMs) without sacrificing their performance. Current methods either sacrifice the quality of the generated text or do not generalize to new inputs. The authors propose a two-stage solution: a small classifier first identifies potentially unsafe prompts, and then a "safe" model is only applied to those prompts. This approach aims to preserve the high-quality outputs of the original LLM on benign prompts while being safe on malicious ones.

The paper focuses on developing an effective safety classifier, experimenting with different architectures and prompting techniques. Surprisingly, a simple classifier using the final hidden state of the last token is found to be effective at detecting unsafe prompts. Further, the paper discovers that safety-related information is present in the hidden states of LLMs at various layers, with implications for potential improvements in classification efficiency. The findings demonstrate the feasibility of enhancing LLM safety without degrading performance and offer intriguing observations on the internal representation of safety information by LLMs.

# Pros and Cons
## Pros
1. **Results:** The results on WJ and MMLU are interesting, showing that the proposed approach is effective in reducing the targeted safety risks.
2. **Layer-Wise Analysis of Models for Safety Assessment:** The layer-wise analysis of models for safety assessment is a good approach to understand the safety of the models, and seems to be newly introduced in this paper.
3. **Quality and Writing:** The paper overall seems to be well-written and fairly easy to read. Addition of more details in the Appendix section could be beneficial.

# Cons
1. **Novelty:** The novelty seems limited, with the previous work having used prompting-based approaches evaluate whether the input prompts are safe or not. The concept of having separate "safe" and "unsafe" models seems to be unique to this paper, however.
2. **Limited Set of Cited Papers:** There seems to be a limited set of cited papers overall. Some relevant papers seem to be missing, and the authors are encouraged to provide a more comprehensive literature review if space permits. The following paper seems to be relevant but on the topic of unlearning: "Thaker, Pratiksha, et al. "Guardrail baselines for unlearning in llms." arXiv preprint arXiv:2403.03329 (2024)."

# Quality and Clarity
The paper overall seems to be of good quality and the concepts are well-explained in spite of the page limit. The authors could extend the Appendix section to provide more details on the experimental setup as well as for a more comprehensive literature review. A clearer presence of the potential future directions could also be beneficial.

# Minor/Major Errors
No major errors were found in the paper. The titles for some of the sections seem to be too long, esp. for section 4.2, even though this is a very minor (potential) issue.

---

### Official Review · Reviewer_r4VS · 2025-03-03
**Review of "Do LLMs Understand The Safety of Their Inputs"**

**Rating:** 6
**Confidence:** 3

**Review:**

Summary:
This paper addresses the issue of performance degradation of LLMs that have been fine-tuned for safety by exploring classifiers that first predict whether a prompt is safe or unsafe, which is then followed by an appropriate LLM selection (i.e., the fine-tuned for safety model is only chosen when a prompt has been classified as unsafe).

Strengths:
- the paper introduces a seemingly effective and easy-to-implement solution to the performance-degradation problem
- the writing in most parts of the paper is clear

Weaknesses:
- only addresses this problem using the Mistral model; does the safe-unsafe classification work just as well for other models?
- no examples or definitions of safe/unsafe prompts. Are the used datasets enough?
- the writing in section 3 is a bit difficult to follow. The purpose of the different training datasets only becomes clear after several read-throughs.

Overall, while this paper introduces an interesting idea, it could benefit from a more rigorous approach of evaluating the feasibility of this idea. At its current state, it feels more like a paper on the Mistral model, rather than for LLMs, more generally.

---

### Decision · Program_Chairs · 2025-03-04

Accept